# Improvement of the In Vitro Cytotoxic Effect on HT-29 Colon Cancer Cells by Combining 5-Fluorouacil and Fluphenazine with Green, Red or Brown Propolis

**DOI:** 10.3390/molecules28083393

**Published:** 2023-04-12

**Authors:** Soraia I. Falcão, Diana Duarte, Moustapha Diallo, Joana Santos, Eduarda Ribeiro, Nuno Vale, Miguel Vilas-Boas

**Affiliations:** 1Centro de Investigação de Montanha (CIMO), Instituto Politécnico de Bragança, Campus de Santa Apolónia, 5300-253 Bragança, Portugal; 2Laboratório Associado para a Sustentabilidade e Tecnologia em Regiões de Montanha (SusTEC), Instituto Politécnico de Bragança, Campus de Santa Apolónia, 5300-253 Bragança, Portugal; 3OncoPharma Research Group, Center for Health Technology and Services Research (CINTESIS), Rua Doutor Plácido da Costa, 4200-450 Porto, Portugal; 4CINTESIS@RISE, Faculty of Medicine, University of Porto, Alameda Professor Hernâni Monteiro, 4200-319 Porto, Portugal; 5Department of Community Medicine, Health Information and Decision (MEDCIDS), Faculty of Medicine, University of Porto, Alameda Professor Hernâni Monteiro, 4200-319 Porto, Portugal

**Keywords:** propolis characterization, phenolic compounds, colorectal cancer, combination effect, antineoplastic drugs

## Abstract

Cancer is regard as one of the key factors of mortality and morbidity in the world. Treatment is mainly based on chemotherapeutic drugs that, when used in targeted therapies, have serious side effects. 5-fluorouracil (5-FU) is a drug commonly used against colorectal cancer (CRC), despite its side effects. Combination of this compound with natural products is a promising source in cancer treatment research. In recent years, propolis has become the subject of intense pharmacological and chemical studies linked to its diverse biological properties. With a complex composition rich in phenolic compounds, propolis is described as showing positive or synergistic interactions with several chemotherapeutic drugs. The present work evaluated the in vitro cytotoxic activity of the most representative propolis types, such as green, red and brown propolis, in combination with chemotherapeutic or CNS drugs on HT-29 colon cancer cell lines. The phenolic composition of the propolis samples was evaluated by LC-DAD-ESI/MS^n^ analysis. According to the type of propolis, the composition varied; green propolis was rich in terpenic phenolic acids and red propolis in polyprenylated benzophenones and isoflavonoids, while brown propolis was composed mainly of flavonoids and phenylpropanoids. Generally, for all propolis types, the results demonstrated that combing propolis with 5-FU and fluphenazine successfully enhances the in vitro cytotoxic activity. For green propolis, the combination demonstrated an enhancement of the in vitro cytotoxic effect compared to green propolis alone, at all concentrations, while for brown propolis, the combination in the concentration of 100 μg/mL gave a lower number of viable cells, even when compared with 5-FU or fluphenazine alone. The same was observed for the red propolis combination, but with a higher reduction in cell viability. The combination index, calculated based on the Chou–Talalay method, suggested that the combination of 5-FU and propolis extracts had a synergic growth inhibitory effect in HT-29 cells, while with fluphenazine, only green and red propolis, at a concentration of 100 μg/mL, presented synergism.

## 1. Introduction

Within the bee products, propolis is the most important “chemical weapon” of bees against pathogen microorganisms, and it has been used as a remedy by humans since ancient times [1]. Propolis is a natural resinous substance produced by honeybees by mixing salivary secretions with plant-based natural materials, such as exudates from leaves, shoots buds, wounds and sap flowers, and employed in the beehives as a building and defensive material [1,2]. With an extremely complex composition, which is strongly dependent on the plant sources available around the hive and on the geographic and climatic conditions, several types of propolis have been characterized [3]. Among them, three propolis types are gaining more relevance within the international market with the establishment of new quality standards [4]. Brown propolis, also known as poplar propolis, with origin mainly in *Populus* spp. trees, is characterized by a composition rich in flavonoids without B-ring substituents (e.g., pinocembrin, pinobanksin, galangin, chrysin) and their esters, together with phenylpropanoids and their esters (e.g., caffeic acid phenylethyl ester, CAPE) [5]. For green propolis, originated from Brazil, the botanical source is the plant *Baccharis dracunculifolia* DC, with the most abundant compounds in the samples being the prenylated phenylpropanoids (e.g., artepillin C, drupanin), caffeoylquinic acids (e.g., dicaffeoylquinic acid) and its derivatives [6]. On the other hand, red propolis, with origin in tropical countries, is described to have its origin in *Dalgergia* spp. and/or *Clusia* sp., with isoflavonoids and polyprenylated benzophenones as the main compounds [7]. The knowledge of the compounds responsible for its bioactivities is linked to the growing use of propolis as a component of pharmaceutics, cosmetics and food supplements with anti-microbial, anti-inflammatory, antiviral, anticarcinogenic and immunomodulatory activities [8,9].

Colorectal cancer (CRC) is the second leading cause of death by cancer in the United States of America, and in 2020, more than 147,000 new diagnosed cases and 53,000 deaths were estimated [10]. Besides surgery, chemotherapy has an important role in the treatment of CRC. Chemotherapeutic drugs act on cancer cell growth and prevent them from spreading to other parts of the body. This type of treatment usually involves the use of antineoplastic drugs in higher dosages, which also affect normal cells and result in severe side effects [11]. 5-fluorouracil (5-FU) is an antineoplastic drug commonly used in the therapy of CRC, but its use is limited by its short half-life, high cytotoxicity and low bioavailability [12]. To overcome these limitations, higher doses and long-term use of 5-FU is necessary, which results in increased side effects. Novel strategies, such as combination therapy, are needed to reduce 5-FU doses and exposure time, while maintaining their anticancer effect [12]. By achieving synergism between the drugs, i.e., reaching a potentiation of the effectiveness compared to the two drugs alone, drug combinations allow decreasing the therapeutical dose, while reducing its side effects [13]. Several studies have indeed demonstrated that drug combination is more effective than monotherapy [14,15,16,17].

Our group has been studying the strategy of combining antineoplastic drugs with several classes of repurposed drugs [18,19]. Drug repurposing is another promising strategy that uses drugs already approved by the FDA for novel therapeutic indications besides the original [20]. This strategy saves time and money, while ensuring the necessary safety and toxicity profiles [21]. Our recent findings revealed that some central nervous system (CNS) drugs have promising cytotoxic profiles against HT-29 colon and MCF-7 breast cancer cells, both alone and combined with antineoplastic drugs [18]. The most promising drug was fluphenazine, a drug that blocks postsynaptic mesolimbic dopaminergic D1 and D2 receptors in the brain and depresses the release of hypothalamic and hypophyseal hormones [22].

The combination of antineoplastic drugs with natural or semi-synthetic compounds has also been demonstrating promising results [23,24,25,26]. The co-administration of an aqueous propolis extract from Brazil significantly increased tumor regression, compared with using 5-FU alone, and significantly ameliorated the cytopenia and cardiotoxicity induced by 5-FU [27,28]. Further, an ethanolic extract of Iran propolis increased the anti-cancer effect of 5-FU by further inhibiting the onset and progression of colorectal cancer [29]. Nevertheless, no one has reported studying the cytotoxic effects of 5-FU or fluphenazine combined with the different propolis types—brown, green and red—in CRC treatment. In this work, we hypothesized that the antineoplastic drug 5-FU and the CNS agent fluphenazine could synergistically act with brown, red and green propolis extracts in HT-29 colon cancer cells. The aim of using these drugs in combination with different propolis samples was to enhance the cytotoxic effect of 5-FU and fluphenazine for colon cancer therapy, exploring recognized standardized propolis types. These in vitro preliminary results demonstrate that propolis extract increased the efficiency of 5-FU and fluphenazine and so, they should be explored in future in vivo studies to confirm its potential as adjunct therapy for colorectal cancer.

## 2. Results and Discussion

### 2.1. Phenolic Compounds Characterization by LC/DAD/ESI-MS^n^

Propolis’ chemical composition is directly influenced by the floral sources available in the beehive surroundings, and it can be quite diverse, taking into account the phenolic compounds, which are the main chemical classes responsible for its bioactivity. For the detailed characterization and quantification of the phenolic profile present in the propolis extracts, the chromatographic methods of LC/DAD/ESI-MS^n^ were employed, which allowed the elucidation of the phenolic compounds through the comparison of their chromatographic behavior, UV spectra and MS information to those of reference standards. In the case of commercial unavailability, the structural information was confirmed with UV information combined with MS fragmentation patterns previously reported in the literature [30]. The phenolic composition of the three propolis types is described in Table 1, Table 2 and Table 3, and the chromatographic profiles can be found in the Appendix A.

The three propolis types displayed different phenolic profiles for their extract, in line with previous reports [5,30,31]. For Brazilian green propolis, 21 phenolic compounds were identified. They consisted of 6 phenolic acids, where *p*-coumaric acid (*m*/*z* 163) and dicaffeoylquinic acid (*m*/*z* 515) were the most representative; 4 flavonoids, which included 3 flavonols—kaempferol (*m*/*z* 285), kaempferide and its isomer (*m*/*z* 299)—and 1 dihydroflavonol, dihydrokaempferide (*m*/*z* 301); and 11 terpenic phenolic compounds, including drupanin (*m*/*z* 231), artepillin C (*m*/*z* 299) and baccharin (*m*/*z* 363), described as chemical markers for this type of propolis, with origin in *B. dracunculifolia*, Table 1, Appendix A [32]. Compounds 15 and 19 were identified in this type of propolis for the first time; however, they require additional confirmation by NMR techniques.

**Table 1 molecules-28-03393-t001:** Characterization of the phenolic compounds in green propolis extract, obtained by LC/DAD/ESI-MS^n^.

Nr	t_R_ (min)	λ_max_ (nm)	[M − H]^−^ *m*/*z*	MS^2^ (% Base Peak)	Compound	mg/g Extract
1	4.9	294sh, 325	353	191 (100), 179 (8), 135 (1)	5-O-Caffeoylquinic acid ^a,b^	1.35 ± 0.04
2	6.8	292, 323	179	135	Caffeic acid ^a,b^	1.18 ± 0.04
3	9.7	310	163	119	*p*-Coumaric acid ^a,b^	9.92 ± 0.01
4	10.8	294sh, 325	515	353	Dicaffeoylquinic acid ^b,c^	6.04 ± 0.40
5	11.8	294sh, 325	515	353	Dicaffeoylquinic acid (isomer) ^b,c^	8.81 ± 0.04
6	18.5	294sh, 325	677	515	Tricaffeoylquinic acid ^b,c^	3.34 ± 0.01
7	24.9	292	301	283 (100), 151 (29)	Dihydrokaempferide ^b,c^	24.53 ± 0.06
8	27.7	267, 365	285	285 (100), 257 (13), 151 (20)	Kaempferol ^a,b^	1.47 ± 0.01
9	30.9	321	247	203	5-Isoprenyl caffeic acid ^b,d^	0.36 ± 0.02
10	36.7	315	231	187	Drupanin ^b,c^	4.90 ± 0.02
11	39.8	310	327	283	Dihydroconiferyl *p*-coumarate ^b,c^	0.44 ± 0.01
12	40.3	315	315	271 (100), 241 (70), 285 (59)	Cappilartimisin A ^b,c,d^	2.03 ± 0.01
13	45.6	315	315	271 (100), 241 (72), 285 (55)	Cappilartimisin A (isomer) ^b,d^	3.72 ± 0.02
14	49.2	266, 365	299	284	Kaempferide ^b,c^	35.66 ± 0.13
15	50.1	266, 365	299	284	Kaempferide (isomer) ^b,c^	14.95 ± 0.02
16	50.5	269, 363	329	314	NI	
17	53.2	316	393	349 (100), 163 (91), 145 (53)	5-Isoprenyl caffeic acid-p-coumaric acid ester ^b,d^	4.05 ± 0.02
18	53.7	319	315	245 (100), 201 (41), 271 (11), 257 (11)	Cappilartimisin A (isomer) ^b,d^	1.57 ± 0.01
19	56.2	315	379	231	Drupanin derivative ^b^	0.68 ± 0.01
20	57.4	284	377	245 (100), 319 (95), 349 (66)	*E*-Baccharin 5″-aldehyde ^b,e^	1.23 ± 0.01
21	61.3	314	299	255	Artepillin C ^b,c^	24.03 ± 0.04
22	62.1	284	363	187	Baccharin ^b,e^	2.10 ± 0.01
23	67.4	282	447	297 (100), 149 (10)	NI	
24	68.2	277, 320	613	511	NI	

^a^ Confirmed with standard; ^b^ Confirmed with MS^n^ fragmentation; ^c^ [30]; ^d^ [33]; ^e^ [34]; NI: non-identified.

For red propolis, 26 phenolic compounds were tentatively identified. They consisted of 4 phenolic acids, including caffeic acid (*m*/*z* 179) and its derivatives, such as caffeic acid isoprenyl ester (*m*/*z* 247); 17 flavonoids, which included 1 dihydroflavonol, pinobanksin-3-*O*-acetate (*m*/*z* 313), 1 flavone, chrysin (*m*/*z* 253), 4 flavanones, such as liquiritigenin (*m*/*z* 255), naringenin (*m*/*z* 271) and pinocembrin (*m*/*z* 255), 8 isoflavonoids, including formononetin (*m*/*z* 267), vestitol and its isomer neovestitol (*m*/*z* 271), 1 chalcone, isoliquiritigenin (*m*/*z* 255), 1 pterocarpan, 3,4-dihydroxy-9-methoxypterocarpan (*m*/*z* 285), 2 flavonoid pigments, retusapurpurin A and B (*m*/*z* 521); 1 triterpene, cycloartenol/α-amyrin/β-amyrin (*m*/*z* 425); and, finally, 3 polyprenylated benzophenones, which included hydroxyguttiferone (*m*/*z* 617), guttiferone E/xantochymol and oblongifolin B (*m*/*z* 601), Table 2 and Appendix A.

The botanical source of the red propolis resin can be attributed to more than one plant: *Dalbergia ecastophyllum*, which is the main source of the isoflavonoids formononetin, biochanin A, vestitol, neovestitol and the flavanone pinocembrin, and *Symphonia globulifera*, belonging to the Clusiaceae family, which is the main source of the polyprenylated benzophenones found in the sample [35]. The relative contribution of each species to the phenolic composition of the resin varies, depending on the geographic location and climatic conditions [35].

**Table 2 molecules-28-03393-t002:** Characterization of the phenolic compounds in red propolis extract, obtained by LC/DAD/ESI-MS^n^.

Nr	t_R_ (min)	λ_max_ (nm)	[M − H]^−^ *m*/*z*	MS^2^ (% Base Peak)	Compound	mg/g Extract
1	6.8	292, 323	179	135	Caffeic acid ^a,b^	0.11 ± 0.00
2	17.4	276, 312	255	135 (100), 119 (10)	Liquiritigenin ^b,c^	5.20 ± 0.06
3	18.4	279, 310	285	270	Vestitone ^b,d^	0.86 ± 0.03
4	19.1	289	283	268	Calycosin ^b,c^	1.83 ± 0.02
5	21.3	276, 309	315	300	Violanone ^b,e^	0.41 ± 0.02
6	22.0	280, 342	285	270 (100), 267 (17), 179 (4)	3,4-Dihydroxy-9-methoxypterocarpan ^b,e^	1.29 ± 0.02
7	23.6	291	271	151	Naringenin ^a,b^	2.64 ± 0.00
8	25.1	280	283	268	Biochanin A ^b,d^	0.98 ± 0.00
9	30.1	281	299	284	Sativanone ^b,f^	1.81 ± 0.02
10	32.7	282	271	227 (100), 109 (86), 135 (83)	Vestitol ^b,d^	26.16 ± 0.02
11	33.3	280, 320	267	252	Formononetin ^b,d^	5.67 ± 0.04
12	33.6	240, 370	255	135 (100), 119 (25)	Isoliquiritigenin ^b,d^	2.42 ± 0.00
13	36.3	282	271	135 (100), 227 (74), 109 (62)	Neovestitol ^b,d^	17.01 ± 0.00
14	39.3	298, 325	247	179 (100), 135 (16)	Caffeic acid isoprenyl ester ^a,b^	20.95 ± 0.05
15	40.9	298, 325	247	179 (100), 135 (16)	Caffeic acid isoprenyl ester (isomer) ^a,b^	0.37 ± 0.01
16	41.7	298, 325	269	178 (100), 135 (96)	Caffeic acid benzyl ester ^b,g^	0.47 ± 0.00
17	43.3	289	255	213 (100), 211 (55), 151 (36)	Pinocembrin ^a,b^	2.14 ± 0.01
18	45.7	268, 313	253	209	Chrysin ^a,b^	1.52 ± 0.01
19	46.3	294	313	253 (100), 271 (20)	Pinobanksin-3-*O*-acetate ^b,g^	1.97 ± 0.01
20	53.5	324	239	197 (100), 135 (36), 148 (19)	7-Hydroxyflavanone ^b,d^	1.02 ± 0.02
21	54.3	283	397	123 (100), 167 (97), 351 (40)	NI	
22	60.5	285, 481	521	397 (100), 491 (45)	Retusapurpurin B ^b,h^	0.47 ± 0.01
23	64.9	284, 481	521	397 (100), 491 (60)	Retusapurpurin A ^b,h^	0.94 ± 0.01
24	67.5	264, 327	425	410 (100), 367 (43), 355 (41)	Cycloartenol/α-amyrin/β-amyrin ^b,h^	
25	81.2	244, 351	617	465 (100), 481 (40), 521 (15)	16-Hydroxyguttiferone ^b,h^	0.02 ± 0.00
26	83.8	244, 351	601	465	Guttiferone E/Xanthochymol ^b,d^	27.95 ± 0.30
27	84.0	244, 351	601	327 (100), 273 (26), 271 (15)	Oblongifolin B ^b,d^	22.13 ± 0,21

^a^ Confirmed with standard; ^b^ Confirmed with MS^n^ fragmentation; ^c^ [36]; ^d^ [31]; ^e^ [7]; ^f^ [37]; ^g^ [2]; ^h^ [38]; NI: non-identified.

The results obtained for brown propolis showed 39 phenolic compounds identified. They consisted of 14 phenolic acids and 25 flavonoids (11 dihydroflavonols, 7 flavonols, 5 flavones and 2 flavanones), Table 3 and Appendix A. Overall, the phenolic composition followed the common profile found in propolis samples from temperate regions, with origin in the *Populus* spp. resins, where the most representative compounds were the phenolic acids and their derivatives, mainly caffeic acid (*m*/*z* 179), caffeic acid isoprenyl ester and its isomer (*m*/*z* 247), caffeic acid benzyl ester (*m*/*z* 269) and the flavonoids, such as chrysin (*m*/*z* 253), galangin (*m*/*z* 269), pinobanksin (*m*/*z* 271), pinocembrin (*m*/*z* 255) and its derivatives, Table 3 [5].

**Table 3 molecules-28-03393-t003:** Characterization of the phenolic compounds in brown propolis extract, obtained by LC/DAD/ESI-MS^n^.

Nr	t_R_ (min)	λ_max_ (nm)	[M − H]^−^ *m*/*z*	MS^2^ (% Base Peak)	Compound	mg/g Extract
1	6.8	292, 323	179	135	Caffeic acid ^a,b^	6.27 ± 0.09
2	9.7	310	163	119	*p*-Coumaric acid ^a,b^	4.84 ± 0.03
3	10.6	295, 322	193	133 (100), 149 (49), 177 (15)	Ferulic acid ^a,b^	1.40 ± 0.01
4	11.1	298, 319	193	133 (100), 149 (49), 177 (15)	Isoferulic acid ^a,b^	5.25 ± 0.09
5	12.8	228	121		Benzoic acid ^a,b^	1,07 ± 0.01
6	15.9	295sh, 322	207	192 (100), 163 (62)	3,4-Dimethyl-caffeic acid ^a,b^	8.25 ± 0.04
7	19.2	287	285	267 (100), 239 (25), 252 (16)	Pinobanksin-5-methyl ether ^b,c^	23.95 ± 0.09
8	21.0	309	177	163 (100), 119 (16)	*p*-Coumaric acid methyl ester ^a,b^	3.22 ± 0.02
9	21.3	256, 355	315	300	Quercetin-3-methyl ether ^b,c^	3.95 ± 0.11
10	23.8	292	271	253 (100), 225 (22), 151 (8)	Pinobanksin ^b,c^	19.79 ± 0.12
11	27.0	269, 337	269	225 (100), 151 (20)	Apigenin ^a,b^	5.06 ± 0.01
12	27.7	267, 365	285	285 (100), 257 (13), 151 (20)	Kaempferol ^a,b^	6.94 ± 0.04
13	29.3	253, 370	315	300	Isorhamnetin ^a,b^	6.63 ± 0.09
14	30.1	267, 352	299	284	Kaempferol-methyl ether ^b,c^	10.05 ± 0.07
15	32.6	311	173	129	Cinnamylidenacetic acid ^b,c^	18.14 ± 0.12
16	35.9	256, 367	315	165	Rhamnetin ^b,c^	2.76 ± 0.10
17	36.5	265, 300sh, 352	283	268 (100), 239 (76)	Galangin-5-methyl ether ^b,c^	3.66 ± 0.02
18	39.3	298, 325	247	179 (100), 135 (16)	Caffeic acid isoprenyl ester ^a,b^	12.49 ± 0.04
19	40.9	298, 325	247	179 (100), 135 (16)	Caffeic acid isoprenyl ester (isomer) ^a,b^	15.45 ± 0.20
20	41.7	298, 325	269	178 (100), 135 (96)	Caffeic acid benzyl ester ^b,c^	16.78 ± 0.03
21	43.3	289	255	213 (100), 211 (55), 151 (36)	Pinocembrin ^a,b^	140.6 ± 0.16
22	44.5	290	285	139 (100), 145 (42)	NI	
23	45.7	268, 313	253	209	Chrysin ^a,b^	66.93 ± 0.21
24	46.4	294	313	253 (100), 271 (20)	Pinobanksin-3-*O*-acetate ^b,c^	105.5 ± 0.05
25	47.1	266, 300sh, 359	269	269 (100), 241 (61)	Galangin ^a,b^	95.17 ± 0.19
26	48.9	268, 331	283	269	Acacetin ^a,b^	4.72 ± 0.01
27	49.6	265, 300sh, 350sh	283	269	6-Methoxychrysin ^b,c^	4.88 ± 0.02
28	51.1	250, 268sh, 343	313	298	Chrysoeriol-methyl ether ^b,c^	5.48 ± 0.01
29	52.0	294, 310	231	163 (100), 119 (12)	*p*-Coumaric isoprenyl ester ^b,c^	4.72 ± 0.04
30	52.6	295, 324	295	178 (100), 135 (60)	Caffeic acid cinnamyl ester ^b,c^	11.74 ± 0.04
31	53.6	289	327	253 (100), 271 (10)	Pinobanksin-3-*O*-propionate ^b,c^	48.64 ± 0.11
32	56.5	289	269	254 (100), 251 (54), 165 (22)	3-Hydroxy-5-methoxyflavanone ^b,c^	11.90 ± 0.02
33	58.2	292	417	297 (100), 402 (85), 267 (67)	Pinobanksin-methyl-ether-3-*O*-phenylpropionate ^b,d^	13.29 ± 0.01
34	59.1	292	475	415	Pinobansin-3-*O*-acetate-5-*O*-hydroxyphenylpropionate ^b,c^	16.84 ± 0.10
35	59.4	308	431	281	NI	
36	59.8	292	417	267 (100), 281 (100)	Pinobanksin-methyl-ether-3-*O*-phenylpropionate (isomer) ^b,d^	8.93 ± 0.04
37	60.3	292	475	415	Pinobansin-3-*O*-acetate-7-*O*-hydroxyphenylpropionate ^b,c^	30.23 ± 0.15
38	60.6	294, 320	413	161	NI	
39	63.7	292	355	253	Pinobanksin-3-*O*-pentanoate or 2-methylbutyrate ^b,c^	15.87 ± 0.10
40	65.1	292, 322	315	179 (100), 135 (31)	Caffeic acid derivative	3.00 ± 0.01
41	65.5	292	403	253 (100), 271 (21)	Pinobanksin-3-*O*-phenylpropionate ^b,c^	10.69 ± 0.01
42	67.0	292	369	253 (100), 271 (14)	Pinobanksin-3-*O*-hexanoate ^b,c^	23.90 ± 0.06

^a^ Confirmed with standard; ^b^ Confirmed with MS^n^ fragmentation; ^c^ [2]; ^d^ [39]; NI: non-identified.

Table 1, Table 2 and Table 3 show the quantification of the phenolic compounds (mg/g extract) in the propolis extracts through LC/DAD/ESI-MS^n^ at 280 nm. While green propolis had high concentrations of the flavonoid kaempferide (35.66 ± 0.13 mg/g), followed by artepillin C (24.03 ± 0.04 mg/g), red propolis had the polyprenylated benzophenones as its main compounds, with values of 27.95 ± 0.30 mg/g and 22.13 ± 0.21mg/g for guttiferone E/xanthochymol and oblongifolin B, respectively, followed by the isoflavans vestitol (26.16 ± 0.02 mg/g) and neovestitol (17.01 ± 0.00 mg/g). Further, a high value of caffeic acid isoprenyl ester was found (20.95 ± 0.05 mg/g). Finally, brown propolis showed a high concentration of the flavanone pinocembrin (140.6 ± 0.16 mg/g), followed by pinobanksin-3-*O*-acetate (105.5 mg/g), galangin (95.17 mg/g) and chrysin (66.93 mg/g). Caffeic acid was the compound common to all the samples, ranging in value from 0.11 mg/g for red propolis to 6.27 mg/g in brown propolis. Further, *p*-coumaric acid and kaempferol were found in both green and brown propolis, while caffeic acid ester derivatives, as well as pinocembrin, chrysin and pinobanksin-3-*O*-acetate, were common to both red and brown propolis, Table 1, Table 2 and Table 3.

Figure 1 shows the main classes of phenolic compounds found in the composition of the three propolis types. The results must be viewed in equivalent terms, since some of the compounds were quantified using an equivalent standard, rather than the specific compound. In terms of the total phenolic compounds, brown propolis showed the highest content. Flavonoids and phenolic acids were found in all the samples, with values in the ranges 16.90–686.3 mg/g and 21.90–112.6 mg/g, respectively. Terpenic phenolic acids were only found in green propolis, while polyprenylated benzophenones, isoflavonoids and terpenoids were exclusive to red propolis.

### 2.2. Evaluation of Propolis Extracts Effect on HT-29 Colon Cancer Cell Viability

The propolis samples were first tested alone in HT-29 cells for 48 h, and cell viability was evaluated by MTT assay. Cells were treated with increasing concentrations of the three samples of propolis, green, red and brown, in a range of concentrations from 6.25 to 100 µg/mL. Our results demonstrate that green propolis had no influence on cell viability in any concentration, while red propolis significantly decreased the viability of HT-29 cells for the concentrations of 50 and 100 µg/mL, Figure 2. Cells incubated with the higher concentrations demonstrate less than 50% of viable cells, and this type of propolis presented the best profile of in vitro cytotoxic effect among all the samples. Brown propolis treatment resulted in a significant reduction of cell viability only for the concentration of 100 µg/mL.

Linking the phenolic content of the extracts with the in vitro cytotoxic results, it seems that the specificity of the phenolic profile is more pertinent than the quantity; although red propolis was not the sample with the highest content of total phenolics, a high concentration of benzophenones and isoflavonoids were found, which were absent in the other propolis types. Dantas Silva et al. 2017 demonstrated that red propolis has great cytotoxic potential on HL-60, HCT-116, OVCAR-8 and SF-295 cancer cell lines when compared with green and brown Brazilian propolis. Polyisoprenylated benzophenone, xanthochymol and isoflavone formononetin have been associated with the cytotoxic activity of red propolis when its fractions were tested separately [40]. Several isoflavonoids from Brazilian red propolis presented a high antioxidant and anti-inflammatory effect, especially formononetin, vestitol and neovestitol [41], present in high quantities in the red propolis sample under study.

### 2.3. Evaluation of 5-FU and Fluphenazine Effect on HT-29 Colon Cancer Cell Viability after Treatment with Propolis Extract

After treatment with propolis samples, the cells were incubated with increasing concentrations (0–100 μM) of 5-FU and fluphenazine for 48 h. Cell viability was evaluated by the MTT assay. The results demonstrate that all concentrations of 5-FU above 10 µM caused a significant reduction of cell viability, with approximately 20% of non-viable cells. Nevertheless, treatment with 5-FU resulted in a plateau of inhibition, and it was found that increasing its concentration up to 100 µM did not improve its in vitro cytotoxic effect. Interestingly, it was found that treatment with 5-FU induced less cytotoxicity than treatment with fluphenazine, Figure 3. Treatment of HT-29 colon cancer cells with increasing concentrations (0–100 µM) of the repurposed fluphenazine resulted in a significant cell viability reduction for the whole range of concentrations, with more than 50% of dead cells in all treatments above 10 µM. The half-inhibitory concentration (IC_50_) values for each compound are summarized in Table 4, and demonstrated IC_50_ values for 5-FU and fluphenazine of 3.79 and 1.86 µM, respectively. Moreover, the IC_50_ values for all propolis samples demonstrated values >100, 53.03 and 56.54 µg/mL for green, red and brown propolis, respectively. Taken together, these results demonstrate that both the antineoplastic drug 5-FU and the repurposed drug fluphenazine are potent against HT-29 colon cancer cells, with fluphenazine having a lower value of IC_50_ than 5-FU, supporting the use of repurposed drugs as anticancer agents in monotherapy. Moreover, it was found that red propolis is the most promising among all the propolis samples, and so, the best candidate for complementary research procedures involving in vivo studies.

### 2.4. Evaluation of the Combination of 5-FU with Propolis Extracts on HT-29 Colon Cancer Cell Viability

After determining the IC_50_ values for each compound/propolis alone, HT-29 cells were treated with the combination of 5-FU or fluphenazine and each propolis extract. This combination model consisted of varying the concentration of each propolis extract while maintaining a fixed concentration (IC_50_ value) of the antineoplastic drug 5-FU or the repurposed drug fluphenazine, Figure 4.

The results regarding the green propolis combination with 5-FU demonstrate an enhancement of the in vitro cytotoxic effect compared to green propolis alone, for all concentrations. Nevertheless, only the results for the combination of 5-FU with the highest concentration of green propolis (100 µg/mL) yielded a statistical difference, with about 50% of viable cells. The combination of red propolis with 5-FU also produced promising results, especially for the combination of 3.78 µM with red propolis at a concentration of 100 μg/mL, where it was found that this drug combination resulted in significantly less viable cells than both 5-FU and red propolis alone at 100 μg/mL. Results regarding the combination of brown propolis with 5-FU also yielded a lower number of viable cells than brown propolis and 5-FU alone, particularly when combining 3.78 µM of 5-FU with 100 μg/mL of brown propolis. Taken together, these results demonstrate that combination with 5-FU successfully enhances the in vitro cytotoxic effect of these different propolis extracts, especially in higher concentrations. Moreover, it was found that the combinations of 5-FU with 100 μg/mL of red and brown propolis resulted in enhanced effects than treatments with standalone agents, which demonstrates that the combination regimens are preferable to monotherapy.

### 2.5. Evaluation of the Combination of Fluphenazine with Propolis Extracts on HT-29 Colon Cancer Cell Viability

To explore the combination of a repurposed drug with propolis extracts, the CNS drug fluphenazine was also included in the study design of these combinations, as this drug was previously studied by our group for cancer therapy, having a promising profile, Figure 5. The combination model was applied as for the previously described 5-FU: maintaining a fixed dose of fluphenazine and a variable dose of each propolis extract. The results are very similar to the ones obtained with 5-FU and demonstrate that all propolis samples have an enhancement of their intrinsic in vitro cytotoxic effect while in combination with fluphenazine. Particularly for the treatment with the highest concentration of each propolis sample, these results demonstrate that combination with fluphenazine is promising, since cell viability is lower than both the repurposed drug and the propolis sample alone, for the concentration of 100 μg/mL. Taken together, these results demonstrate that, under in vitro conditions, not only the antineoplastic drug 5-FU, but also the repurposed drug fluphenazine, can be explored in combination regimens with propolis extracts, enhancing the effects of the synthetic compounds against cancer cells. 

The use of repurposed drugs is advantageous over antineoplastic drugs since re-purposed drugs can significantly reduce the cost and time associated with drug development. As these drugs have already been approved for other indications, much of the preclinical and early-stage clinical testing has already been completed. This allows for a more streamlined development process, which ultimately benefits patients by bringing new treatments to market more quickly and at a lower cost. Moreover, repurposed drugs can provide patients with additional treatment options, especially in cases where traditional chemotherapy has failed or is not effective. Finally, using repurposed drugs, such as fluphenazine, in combined treatments can potentially reduce side effects associated with cancer treatment. Since many drugs that have already been approved for other indications, they have established safety profiles and are known to be tolerated well by patients. By repurposing these drugs for cancer therapy, clinicians may be able to avoid some of the toxic side effects associated with traditional chemotherapy.

### 2.6. Evaluation of Drug Interaction in the Combinations of 5-FU/Fluphenazine with Propolis Extracts on HT-29 Colon Cancer Cells

To evaluated the drug interaction for each combination previously tested by MTT assay, we used the CompuSyn software—version 1.0, based on the Chou–Talalay method. As shown in Figure 6, the CI values of 5-FU and propolis extracts in combination were mostly lower than one, suggesting that the growth inhibitory effect of these compounds in combination was mostly synergic in HT-29 cells.

Regarding the combination of fluphenazine and propolis extracts, it was found that only two pairs present synergism, and most of them to be antagonist. In the combination of 5-FU with propolis samples, it was found that the blends with 6.25 and 100 µg/mL of green propolis, 6.25, 12.5, 50 and 100 µg/mL of red propolis, as well as 6.25, 12.5 and 100 µg/mL of brown propolis, to have CI under 1, indicative of synergism. These results demonstrate that the combination of these propolis extracts with 5-FU presents higher combined efficacy than with fluphenazine, revealing that the antineoplastic drug should have an important role in the success of the combination. These results are summarized in Table 5.

The determination of drug synergism is important in drug combinations since it can help enhancing the therapeutic efficacy of the drugs, as it allows for a lower dose of each drug to be administered while still achieving a better therapeutic outcome, reducing the risk of toxicity associated with high doses of individual drugs and improve patient compliance. Moreover, drug combinations that exhibit synergistic effects can lead to the discovery of new drug targets and mechanisms of action, facilitating a better understanding of the underlying disease processes and the molecular pathways involved and helping to identify new targets for drug development and provide insights into the pathophysiology of the disease. Moreover, drug combinations that show synergistic effects can help overcome drug resistance, improving treatment outcomes. Finally, the synergistic interaction between drugs in a combination regimen can improve the safety profile of the treatment, since as lower dose of each drug are required to achieve the same therapeutic effect, the risk of adverse drug reactions and toxicities is reduced. This can improve patient safety and reduce the burden on the healthcare system.

## 3. Materials and Methods

### 3.1. Standards and Reagents

The phenolic compounds, caffeic acid, p-coumaric acid, pinocembrin and chrysin were purchased from Sigma Chemical Co. (St Louis, MO, USA), while genistein and kaempferol were purchased from Extrasynthese (Genay, France). HPLC-grade ethanol and acetonitrile were purchased from Fisher Scientific (Leicester, UK). Water was treated in a Milli-Q water purification system (TGI Pure Water Systems, Houston, TX, USA).

For the cell culture, McCoy’s 5A Medium powder (modified with L-glutamine, without sodium bicarbonate), fetal bovine serum (FBS) and penicillin–streptomycin solution were purchased from Millipore Sigma (Merck KGaA, Darmstadt, Germany). Other cell culture reagents were purchased from Gibco (Thermo Fisher Scientific, Inc., Waltham, MA, USA). Thiazolyl Blue Tetrazolium Bromide (MTT, cat. no. M5655), 5-fluorouracil (5-FU, cat. no. F6627) and fluphenazine dihydrochloride (cat. no. F4765) were obtained from Sigma-Aldrich (Merck KGaA, Darmstadt, Germany).

### 3.2. Propolis Samples

The raw propolis samples had different origins: the poplar propolis was from Portugal, collected in 2020 and supplied by Iberiensis, Lda (Bragança, Portugal), while the green and red propolis were from Brazil, collected in 2019 and supplied by Bee Propolis Brasil (Bambui, Minas Gerais, Brazil). The samples were kept at −20 °C until further analysis.

### 3.3. Phenolic Compounds Extraction

The propolis extracts were obtained through a hydro–ethanolic extraction procedure, as previously described [5]. Approximately 1 g of sample was mixed with 10 mL of 80% of ethanol/water and kept in a water bath at 70 °C and 60 rpm for 1 h. The resulting mixture was filtered and re-extracted in the same conditions. Finally, the resulting extracts were combined, concentrated and freeze-dried.

### 3.4. Phenolic Compounds Characterization by LC/DAD/ESI-MS^n^

The LC/DAD/ESI-MS^n^ analyses were performed on a Dionex Ultimate 3000 UPLC instrument (Thermo Scientific, San Jose, CA, USA) equipped with a diode-array detector and coupled to a mass detector. The column used for high-performance liquid chromatography (HPLC) was a Macherey-Nagel Nucleosil C18 (250 mm × 4 mm id; 5 mm particle diameter, end-capped), and its temperature was maintained at 30 °C. The LC conditions used followed our previous work [2]. The flow rate applied was 1 mL/min, and the injection volume was 10 µL. Spectral data for all peaks were accumulated in the range of 190–600 nm.

The mass spectrometer was operated in the negative ion mode using a Linear Ion Trap LTQ XL mass spectrometer (Thermo Scientific, CA, USA) equipped with an electrospray ionization (ESI) source. The ESI source parameters were as follows: source voltage, 5 kV; capillary voltage, −20 V; tube lens voltage, −65 V; capillary temperature, 325 °C; and sheath and auxiliary gas flow (N_2_) set as 50 and 10 (arbitrary units), respectively. Mass spectra were acquired on full-range acquisition covering 100–1000 *m*/*z*. For the fragmentation study, a data-dependent scan was performed by deploying collision-induced dissociation (CID). The normalized collision energy of the CID cell was set at 35 (arbitrary units). Data acquisition was carried out with the Xcalibur^®^ data system (Thermo Scientific, San Jose, CA, USA). The elucidation of the phenolic compounds was achieved by comparison of their chromatographic behavior, UV spectra and MS information to those of the reference compounds. When standards were not available, the structural information was confirmed with UV data, combined with MS fragmentation patterns previously reported in the literature. Quantification was achieved using calibration curves for caffeic acid (0.002–0.4 mg/mL; y = 6 × 10^7^ × −26,360; *R*^2^ = 0.996), *p*-coumaric acid (0.02–0.4 mg/mL; y = 9 × 10^6^ × −35,105; *R*^2^ = 0.999), genistein (0.0375–0.8 mg/mL; y = 1 × 10^6^ × +48,333; *R*^2^ = 0.999), kaempferol (0.075–1.6 mg/mL; y = 1 × 10^6^ × −5867; *R*^2^ = 0.997), pinocembrin (0.0375–0.8 mg/mL; y = 2 × 10^6^ × +5250; *R*^2^ = 0.997) and chrysin (0.0375–0.8 mg/mL; y = 4 × 10^6^ × −18,959; *R*^2^ = 0.999). When the standard was not available, the compound quantification was expressed in equivalent terms of the structurally closest compound. The assays were performed in duplicate and the results expressed as mg/g of extract.

### 3.5. Cell Culture

#### 3.5.1. Cell Line and Cell Culture

The human colon cancer HT-29 cell line was obtained from the American Type Culture Collection (ATCC; Manassas, VA, USA) and maintained according to ATCC’s recommendations at 37 °C and 5% CO_2_ in McCoy’s 5a Medium Modified supplemented with 10% fetal bovine serum, 100 U/mL penicillin G and 100 µg/mL streptomycin. Cells were maintained in the logarithmic growth phase, and media were changed every 3 days. Cells were trypsinized with 0.25% trypsin-EDTA and subcultured in the same media. HT-29 cells (15,000 cells/well) were seeded in 96-well plates and allowed to adhere overnight before the treatments.

#### 3.5.2. Drug Treatment

The effect of 5-FU, fluphenazine and propolis extracts, for single and combination studies, was evaluated after 48 h of treatment. First, the half-maximal inhibitory concentration (IC_50_) value was determined for 5-FU and fluphenazine alone in HT-29 cells. The 5-FU and fluphenazine concentrations ranged from 0.1 to 100 µM for the single-drug treatment. Propolis extracts were tested alone in concentrations of 6.25, 12.5, 25, 50 and 100 µg/mL [42]. Additionally, combination studies were performed by combining 5-FU or fluphenazine (Drug 1) at their IC_50_ value with the different propolis extracts at the same concentrations tested for the single treatment (Drug 2). The Drug 1 concentration was fixed at the IC_50_ value, and the Drug 2 concentrations were variable. The combined effects of non-equipotent concentrations (non-fixed ratio) were evaluated by MTT assay.

#### 3.5.3. Cell Viability Assay

To determine the effects of 5-FU, fluphenazine and different propolis extracts on the viability of HT-29 cells, an MTT assay was used. For the MTT protocol, after drug treatment, the cell medium was removed, and 200 µL/well of MTT solution (0.5 mg/mL in PBS) was added. Cells were incubated for 3 h, protected from light. After this period, the MTT solution was removed, and DMSO (200 µL/well) was added to solubilize the formazan crystals. Absorbance was measured at 570 nm in an automated microplate reader (Tecan Infinite M200, Tecan Group Ltd., Männedorf, Switzerland). The IC_50_ was determined as each drug concentration showing 50% cell growth inhibition as compared with the control. All conditions were performed three times independently, in triplicate.

#### 3.5.4. Data Analysis

GraphPad Prism 8 (GraphPad Software Inc., San Diego, CA, USA) was used to produce concentration–response curves by nonlinear regression analysis. The viability of cells treated with each drug was normalized to the viability of control cells, and cell viability fractions were plotted versus drug concentrations in the logarithmic scale.

#### 3.5.5. Analysis of Drug Interactions

To quantify the drug interaction between 5-FU/fluphenazine and each propolis extract, the Combination Index (CI) was estimated by the unified theory, introduced by Chou and Talalay [43], using CompuSyn software (version 1.0; ComboSyn, Inc., Paramus, NY, USA). The two drugs were combined in a non-fixed ratio of doses, with a fixed concentration of Drug 1 and variable concentrations of Drug 2. The CI was plotted on the y-axis as a function of the effect level (Fa) on the x-axis to assess the drug synergism between the drug combinations. The CI is a quantitative representation of pharmacological interactions. CI < 1 indicates synergism, CI = 1 indicates an additive interaction and CI > 1 indicates antagonism. The experiments were conducted in triplicate (*n* = 3), with 3 replications at each drug concentration.

#### 3.5.6. Statistical Analysis

The results are presented as mean ± SEM for the n experiments performed. All data were assayed in three independent experiences, in triplicate. Statistical comparisons between the control and treatment groups, at the same time point, were performed with the Student’s *t*-test and one-way ANOVA test. Statistical significance was accepted at *p* values < 0.05.

## 4. Conclusions

The wide range of propolis applications in modern medicine is mainly attributed to the phenolic compounds, which exhibited broad-spectrum biological and pharmacological activities. The phenolic composition is strongly dependent on the botanical sources of the resin. Through LC/DAD/ESI-MS^n^, it was possible to identify and quantify the phenolic compounds in the most common types of propolis produced worldwide. Green propolis showed a high concentration of terpenic phenolic acids, while red propolis presented as main compounds polyprenylated benzophenones and isoflavonoids. Finally, brown propolis revealed a high quantity of flavonoids and phenylpropanoids. Overall, the results demonstrated that the combination of propolis with 5-FU successfully enhances the in vitro cytotoxic effect of these different propolis extracts on HT-29 cancer cells, especially in the higher concentrations tested. The combination of green propolis with 5-FU showed an enhancement of the in vitro cytotoxic effect compared to green propolis alone, for all concentrations. For brown propolis, the combination with 5-FU also resulted in a lower number of viable cells than the propolis extract alone, particularly in the concentration of 100 μg/mL. The combination of red propolis with drugs yielded promising results, namely, for the concentration of 100 μg/mL, where the combination resulted in significantly fewer viable cells than both 5-FU and red propolis alone at 100 μg/mL. Finally, the combination of the propolis samples with fluphenazine showed an improvement of their intrinsic in vitro cytotoxic effect against HT-29 cancer cells.

The findings of this study offer promising perspectives for the use of propolis in combination with chemotherapeutic or CNS drugs in repurposing drugs for cancer treatment. With its complex composition rich in phenolic compounds, propolis has shown in vitro positive or synergistic interactions with several chemotherapeutic drugs, including 5-FU and fluphenazine. The variation in phenolic composition between the different types of propolis suggests that different types may be more effective in combination with different drugs. The synergistic growth inhibitory effect of the combination of 5-FU and propolis extracts in HT-29 cells suggests that further research into the mechanisms of this interaction could lead to the development of more effective drug combination with fewer side effects. Overall, these results provide exciting perspectives for the future use of propolis in cancer treatment, with the potential to improve patient outcomes and reduce the burden of chemotherapy-related side effects. To accomplish this goal, further assays are required to confirm the anticancer potential of these drug combinations, widening the in vitro studies to other cell lines but also upscaling the study to in vivo models.

## Figures and Tables

**Figure 1 molecules-28-03393-f001:**
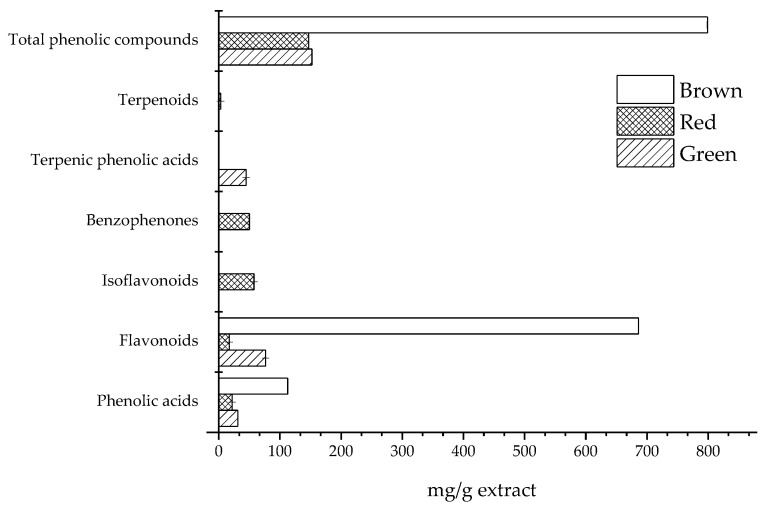
Main classes of phenolic compounds found in green, red and brown propolis.

**Figure 2 molecules-28-03393-f002:**
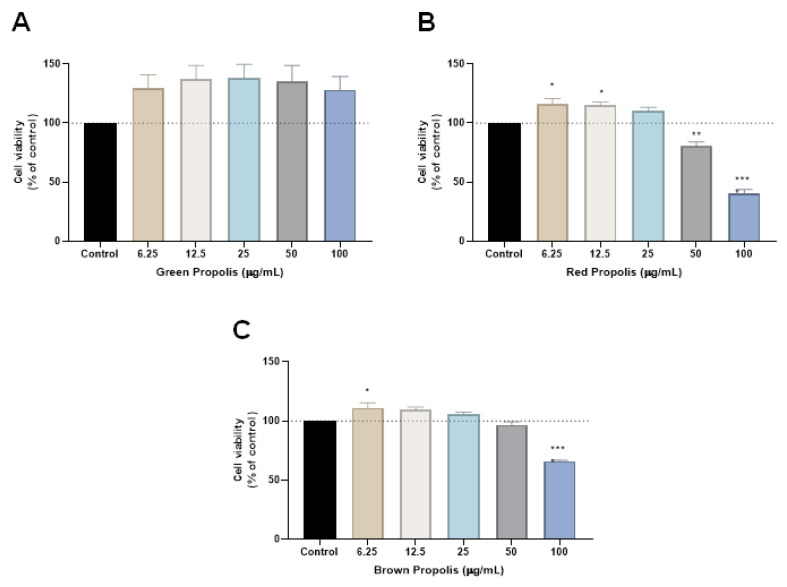
Cell viability of HT-29 colon cancer cells treated with (**A**) green propolis, (**B**) red propolis and (**C**) brown extracts alone. Cultured cells were seeded in 96-well plates and exposed to each extract (6.25–100 μg/mL) for 48 h. Cell viabilities were determined after the final treatment by MTT assay. Each point represents the mean ± SEM relative to the control untreated cells. * Statistically significant vs. control at *p* < 0.05. ** statistically significant vs. control at *p* < 0.01. *** statistically significant vs. control at *p* < 0.001.

**Figure 3 molecules-28-03393-f003:**
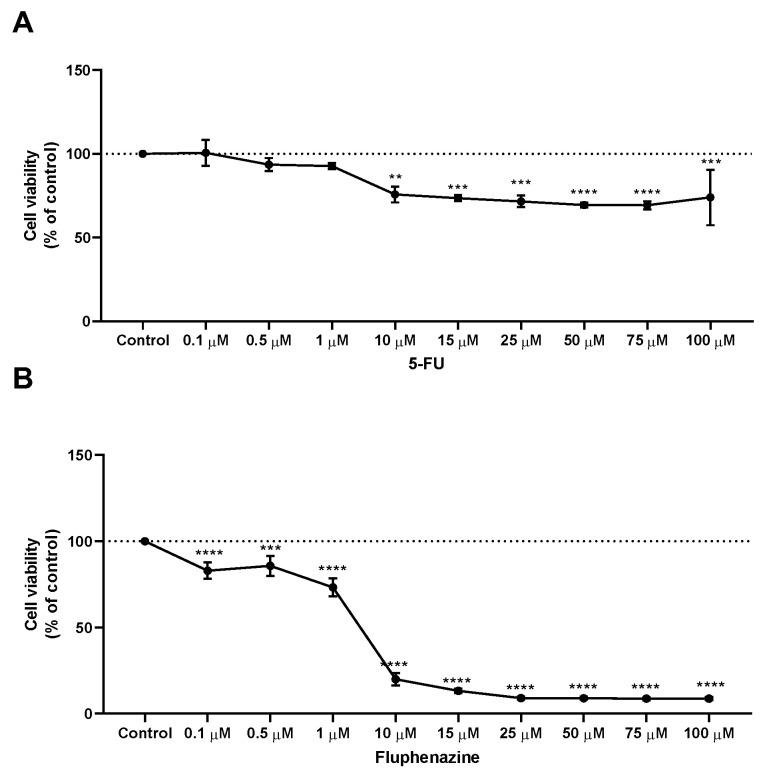
Cell viability of HT-29 colon cancer cells treated with (**A**) 5-FU and (**B**) fluphenazine. Cultured cells were seeded in 96-well plates and exposed to each extract (0.1–100 μM) for 48 h. Cell viabilities were determined after the final treatment by MTT assay. Each point represents the mean ± SEM relative to the control untreated cells. ** statistically significant vs. control at *p* < 0.01. *** statistically significant vs. control at *p* < 0.001. **** statistically significant vs. control at *p* < 0.0001.

**Figure 4 molecules-28-03393-f004:**
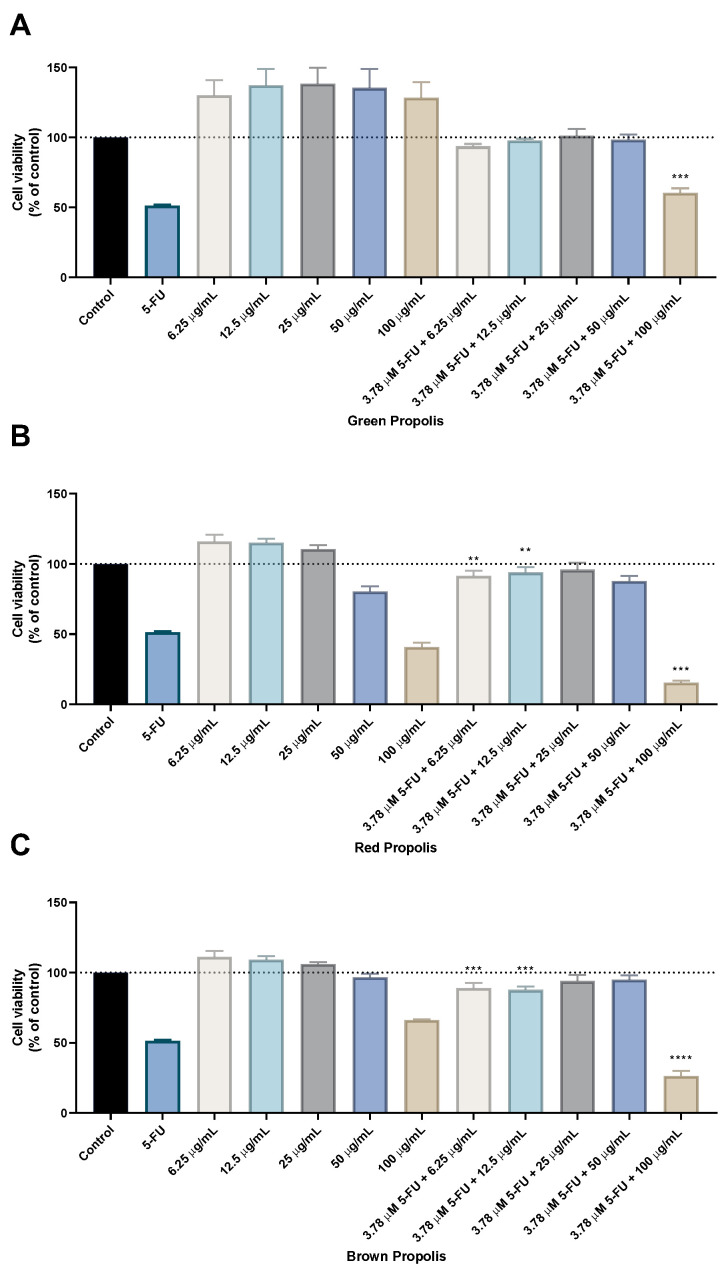
Cell viability of HT-29 colon cancer cells treated with the combination of different propolis extracts and 5-FU. Cultured cells were seeded in 96-well plates and exposed to increasing concentrations of each propolis extract combined with the IC_50_ value of 5-FU (3.78 µM) for 48 h. Cell viabilities were determined after the final treatment by MTT. The drugs in combination were co-administered at the same time. (**A**) The effect of 5-FU plus green propolis on cell viability. (**B**) The effect of 5-FU plus red propolis on cell viability. (**C**) The effect of 5-FU plus brown propolis on cell viability. Each point represents the mean ± SEM relative to the control untreated cells. ** statistically significant vs. control at *p* < 0.01. *** statistically significant vs. control at *p* < 0.001. **** statistically significant vs. control at *p* < 0.0001.

**Figure 5 molecules-28-03393-f005:**
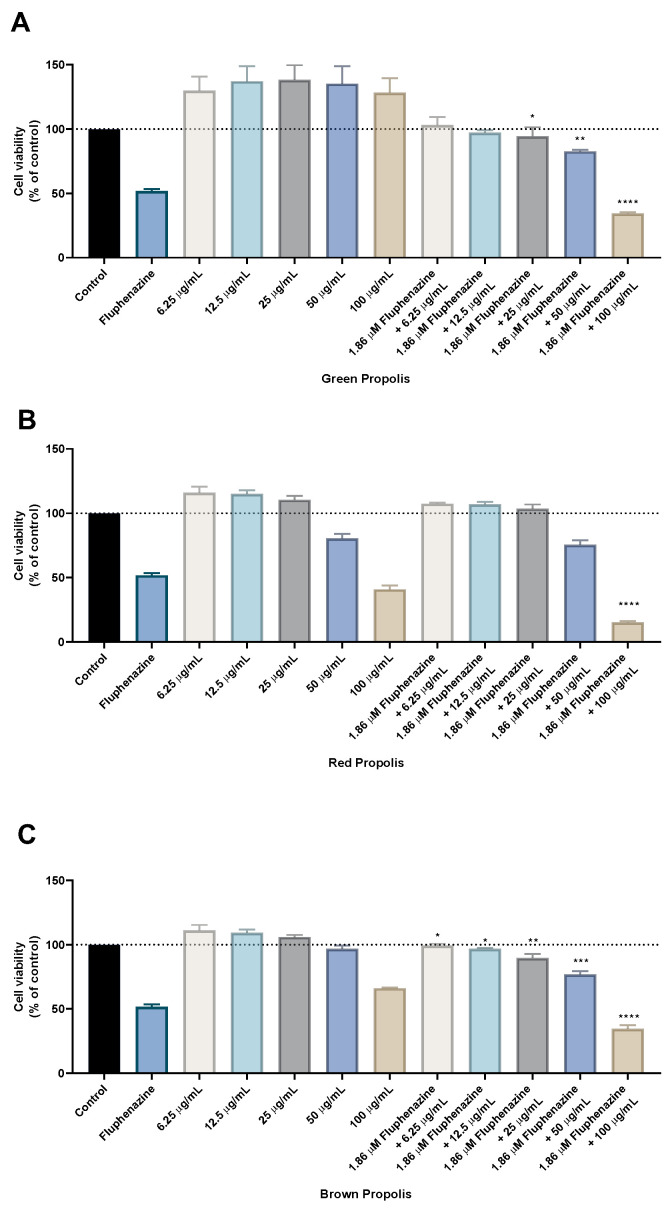
Cell viability of HT-29 colon cancer cells treated with the combination of different propolis extracts and fluphenazine. Cultured cells were seeded in 96-well plates and exposed to increasing concentrations of each propolis extract combined with the IC_50_ value of fluphenazine (1.86 µM) for 48 h. Cell viabilities were determined after the final treatment by MTT. The drugs in combination were co-administered at the same time. (**A**) The effect of fluphenazine plus green propolis on cell viability. (**B**) The effect of fluphenazine plus red propolis on cell viability. (**C**) The effect of fluphenazine plus brown propolis on cell viability. Each point represents the mean ± SEM relative to the control untreated cells. * statistically significant vs. control at *p* < 0.05. ** statistically significant vs. control at *p* < 0.01. *** statistically significant vs. control at *p* < 0.001. **** statistically significant vs. control at *p* < 0.0001.

**Figure 6 molecules-28-03393-f006:**
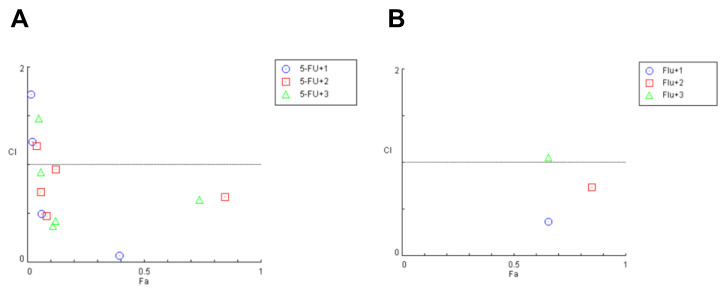
Fa-CI plot of combined treatments of (**A**) 5-FU and (**B**) fluphenazine combined with (1) green propolis, (2) red propolis and (3) brown propolis on HT-29 colon cancer cells. The combination index was calculated using CompuSyn software. CI < 1, =1, and >1 indicates synergistic, additive, and antagonistic effects, respectively.

**Table 4 molecules-28-03393-t004:** Individual IC_50_ of 5-FU, fluphenazine and propolis in HT-29 colon cancer cells.

Drug	IC_50_
5-FU	3.79 µM
Fluphenazine	1.86 µM
Green Propolis	>100 µg/mL
Red Propolis	53.03 µg/mL
Brown Propolis	56.54 µg/mL

**Table 5 molecules-28-03393-t005:** Nature of drug interactions in HT-29 colon cancer cells treated with fluphenazine and 5-FU combined with green, red and brown propolis.

Drug A	Dose A (µM)	Sample B	Dose B (µg/mL)	Effect (Fa)	CI Value	Interaction
Fluphenazine	1.86	Green Propolis	6.25	0.00001	>100	Antagonism
			12.5	0.0289	>100	Antagonism
			25	0.0555	46.59	Antagonism
			50	0.1733	7.92	Antagonism
			100	0.6569	0.36	Synergism
		Red Propolis	6.25	0.00001	>100	Antagonism
			12.5	0.0001	>100	Antagonism
			25	0.001	>100	Antagonism
			50	0.2458	4.86	Antagonism
			100	0.849	0.73	Synergism
		Brown Propolis	6.25	0.0076	>100	Antagonism
			12.5	0.0316	>100	Antagonism
			25	0.1036	18.50	Antagonism
			50	0.2319	5.31	Antagonism
			100	0.6549	1.05	Antagonism
5-FU	3.78	Green Propolis	6.25	0.0635	0.49	Synergism
			12.5	0.0236	1.23	Antagonism
			25	0.00001	>100	Antagonism
			50	0.0165	1.71	Antagonism
			100	0.3949	0.06	Synergism
		Red Propolis	6.25	0.0842	0.47	Synergism
			12.5	0.0606	0.72	Synergism
			25	0.0409	1.19	Additivity
			50	0.1236	0.95	Synergism
			100	0.8457	0.66	Synergism
		Brown Propolis	6.25	0.1100	0.37	Synergism
			12.5	0.1238	0.42	Synergism
			25	0.0606	0.92	Additivity
			50	0.0496	1.47	Antagonism
			100	0.7379	0.64	Synergism

## Data Availability

Data is contained within the article or Appendix A.

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
