# Peer review of "Improvement of the In Vitro Cytotoxic Effect on HT-29 Colon Cancer Cells by Combining 5-Fluorouacil and Fluphenazine with Green, Red or Brown Propolis"

_molecules, 2023, doi:10.3390/molecules28083393_

Round 1
Reviewer 1 Report
Respected,
The manuscript of title "Green, red and brown propolis: improvement of the cytotoxic effect of 5-Fluorouacil in combination with CNS drugs on HT-29 colon cancer cells" presents a very interesting topic and study. In the submitted manuscript, the authors conducted research with the aim of first examining the individual action of extracts of three types of propolis (green, red and brown) in different concentrations from 6.25 to 100 μg/mL, as well as the individual action of two drugs (5-fluorouracil/5-FU and fluphenazine) in concentrations ranged from 0.1 to 100 μM on HT-29 human colorectal cancer cells. Further, the authors studied the effect of combining drugs at their IC50 value in combination with different propolis extracts in the same concentrations tested for individual treatment. The reviewed study provides new knowledge about the influence of the combination of extracts of different types of propolis with drugs such as 5-FU and fluphenazine on the antitumor activity tested on HT-29 human colon cancer cell culture. The scope of the research study and its significance, as well as the presentation of the obtained results, are suitable for publication in the journal Molecules. However, there are some minor corrections that need to be further improved before publication.
In the Abstract part:
L43 MCF-7 cells must be replaced with HT-29 cells
In the Results and Discussion part:
L255 In Figure 4/A, the maximum value on the epsilon axis (y) should be equal to the maximum value specified in Figure 4/B and C (150). The name of Figure 4/C should be changed to Brown propolis.
L288 In Figure 4/A, the maximum value on the epsilon axis (y) should be equal to the maximum value specified in Figure 4/B and C (150). The name of Figure 4/C should be changed to Brown propolis.
L305 MCF-7 cells must be replaced with HT-29 cells
L306 On graph 6/A and B, it is necessary to display the value 1 on the epsilon axis, in the range of values (0 to 2). Figure 6/A is not clearly displayed for interpretation and needs to be rearranged
L320 In the title of Table 5., it is necessary to specify brown propolis instead of “poplar”
Author Response
We would like to thank the reviewers for their valuable comments, suggestions, and contributions to this study. Below you can find the specific actions regarding each comment:
1. In the Abstract part: L43 MCF-7 cells must be replaced with HT-29 cells.
Our response: Changes were made according to the reviewer suggestion.
- In the Results and Discussion part:
L255 In Figure 4/A, the maximum value on the epsilon axis (y) should be equal to the maximum value specified in Figure 4/B and C (150). The name of Figure 4/C should be changed to Brown propolis
Our response: Figure 4 was revised according to the suggestions.
- In the Results and Discussion part:
L288 In Figure 5/A, the maximum value on the epsilon axis (y) should be equal to the maximum value specified in Figure 5/B and C (150). The name of Figure 5/C should be changed to Brown propolis.
Our response: Figure 5 was revised according to the suggestions.
- In the Results and Discussion part:
L305 MCF-7 cells must be replaced with HT-29 cells
Our response: The text was revised according to the reviewer comment.
- In the Results and Discussion part:
L306 On graph 6/A and B, it is necessary to display the value 1 on the epsilon axis, in the range of values (0 to 2). Figure 6/A is not clearly displayed for interpretation and needs to be rearranged.
Our response: We appreciate your feedback and comments on the graphical representation of our results in Figure 6. We would like to clarify that the graphs in Figure 6 were automatically generated by the CompuSyn program. Unfortunately, we are unable to make adjustments to the axes of the graphs or to the way the results are presented. The program has certain limitations in terms of its output format and customization options. We understand that the graphical representation of data is an important aspect of scientific communication. However, in this case, we believe that the automatic generation of the graphs by CompuSyn provides a standardized and objective approach to presenting our results. We hope that this explanation has addressed your concerns regarding the graphical representation in Figure 6.
- In the Results and Discussion part:
L320 In the title of Table 5., it is necessary to specify brown propolis instead of “poplar”.
Our response: Corrections were made according to the reviewer suggestion.
Reviewer 2 Report
In this manuscript, authors reported the “Green, red and brown propolis: improvement of the cytotoxic effect of 5-fluorouacil in combination with CNS drugs on HT-29 colon cancer cells”. Although all experiments were carefully performed and conducted to draw the conclusion, there are several problems and lacks as following:
1. In the title, 5-Fluorouacil à 5-fluorouacil
2. Authors should replace “botanical origin” with “characterization of propolis” in the keywords.
3. In the keywords, drug combination à combination effect
4. Authors should remove “drug repurposing” in the keywords because they have already reported drug repurposing as fluphenazine (Int. J. Mol. Sci. 2021, 22, 7408).
5. DAD and ESI-MS data of peaks 22 and 23 in the Table 2 look almost identical. How did authors distinguish retusapurpurin A and B?
6. In the 26 peak line of the Table 2, Guttiferone E E/Xanthochymol à Guttiferone E/Xanthochymol
7. In the line for peak 37 in the Table 3, what is an isomer of this compound? If it is not an isomer, please remove “(isomer)”.
8. In Figure 1, how did the author quantify the total amount of each component group? If there are no standard compounds, it is recorded that similar compounds were used for quantification, but this will not be able to predict the accuracy of the data.
9. In Tables 1~3, it is necessary to indicate if substances not reported in previous studies were detected. If authors reconfirmed substances already reported in the three types of propolis, this does not correspond to the characterization of propolis.
10. In the Table 5, spaces are required between green propolis 100 μg/mL and red propolis 6.25 μg/mL and between red propolis 100 μg/mL and brown propolis 6.25 μg/mL.
11. In the C of figures 4 and 5, Poplar à Brown Propolis
12. In the Figure 3A, 5-FU does not have an IC50 value up to a concentration of 100 μM, but in the Table 4, the IC50 value of 5-FU is recorded as 3.78 μM. Authors should check and correct which one is right.
13. In A, B, and C of figures 4 and 5, it is desirable to keep the order of the same color for the bar graphs (for the same sample condition, use the bar graph of the same color).
14. I suggest to reduce the concentration of 5-FU and fluphenazine to confirm the combination effect.
15. In the figures 4 and 5, the doses of 5-FU and fluphenazine must be displayed, and authors also add that into figure legends.
16. Authors should add propolis in the subtitle, “2.3. Evaluation of 5-FU and fluphenazine effect on HT-29 colon cancer cell viability”.
17. In the lines 232 and 236, uM à μM
Author Response
We would like to thank the reviewers for their valuable comments, suggestions, and contributions to this study. Below you can find the specific actions regarding each comment:
Reviewer #2:
- In the title, 5-Fluorouacil à 5-fluorouacil.
Our response: The title was revised according to the reviewer suggestion.
- Authors should replace “botanical origin” with “characterization of propolis” in the keywords.
Our response: Specific corrections were made in the manuscript combining the first keyword with the reviewer suggestion.
- In the keywords, drug combination à combination effect.
Our response: Changes were made according to the reviewer suggestion.
- Authors should remove “drug repurposing” in the keywords because they have already reported drug repurposing as fluphenazine (Int. J. Mol. Sci. 2021, 22, 7408).
Our response: The keyword was removed according to the suggestion.
- DAD and ESI-MS data of peaks 22 and 23 in the Table 2 look almost identical. How did authors distinguish retusapurpurin A and B?
Our response: Thanks for the reviewer's comment. The UV data and MS fragmentation for these two compounds are similar. In the absence of standards, we use the information available in the literature and we attributed the designation of B and A according to the order of retention time given that in literature, using similar chromatographic conditions: retusapurpurin B as a lower retention time than retusapurpurin A. Picinelli et al. (2011), observed that the two compounds presented the same UV and MS profile and the confirmation of the two was made through RMN analysis.
References: Anna Lisa Piccinelli, Cinzia Lotti, Luca Campone, Osmany Cuesta-Rubio, Mercedes Campo Fernandez, and Luca Rastrelli, Journal of Agricultural and Food Chemistry 2011 59 (12), 6484-6491, DOI: 10.1021/jf201280z.
- In the 26 peak line of the Table 2, Guttiferone E E/Xanthochymol à Guttiferone E/Xanthochymol.
Our response: Changes were made according to the reviewer suggestion.
- In the line for peak 37 in the Table 3, what is an isomer of this compound? If it is not an isomer, please remove “(isomer)”.
Our response: The table was revised according to the reviewer suggestion.
- In Figure 1, how did the author quantify the total amount of each component group? If there are no standard compounds, it is recorded that similar compounds were used for quantification, but this will not be able to predict the accuracy of the data.
Our response: As stated, when the standard was not available, we used a similar compound (from the same structural family) to quantified. Indeed, the value obtained in those situations is in equivalent terms (a comparative quantification). This approach is frequently used in the characterization of complex matrix such as propolis, and also for example, typical of other spectrophotometrical methods as well, such as quantification of total phenolics by Folin-Cioacalteau. For the quantification of each group value, like is displayed in Figure 1, we expressed the sum of the compounds obtained as mention before, and so, again, the result is in equivalent terms. We add a specific sentence on the description of figure 1 (results) to highlight this aspect.
- In Tables 1~3, it is necessary to indicate if substances not reported in previous studies were detected. If authors reconfirmed substances already reported in the three types of propolis, this does not correspond to the characterization of propolis.
Our response: Indeed, for green propolis, Table 1, there are two compounds, an isomer of kaempferide and a drupanin derivative, never described before. We improved the text identifying this fact. However, we must highlight that propolis is known to be a complex matrix where the chemical composition is significantly dependent on the main botanical origin, but also on the complementary floral sources around the beehive. So, despite the compounds mentioned within tables 1-3 have already been described for some propolis, does not mean that they are necessarily present in all samples. Because of that, when performing assays for biological properties evaluation, it is important to accompanied it with the chemical profile in bioactive compounds.
- In the Table 5, spaces are required between green propolis 100 μg/mL and red propolis 6.25 μg/mL and between red propolis 100 μg/mL and brown propolis 6.25 μg/mL.
Our response: We added specific line boarders to clarify the separation between samples.
- In the C of figures 4 and 5, Poplar à Brown Propolis
Our response: Thank you for pointing this out. Figures 4 and 5 were changed accordingly.
- In the Figure 3A, 5-FU does not have an IC50 value up to a concentration of 100 μM, but in the Table 4, the IC50 value of 5-FU is recorded as 3.78 μM. Authors should check and correct which one is right.
Our response: Thank you for mentioning this. Indeed, when evaluating the cytotoxic effect of 5-FU in this cell line in four different experiments, we have found out that 5-FU reached a plateau of inhibition and that increasing its concentration would not improve its cytotoxic effect. All IC50 values used throughout the manuscript were obtained by curve plots using the GraphPad software. Also, we would like to mention that differences between results from MTT assays in Figure 3A and the obtained IC50 value in Table 4 can be explained by the difference between relative and absolute IC50 values. The values represented in our manuscript are relative IC50, which is the most common IC50 definition. This value represents the concentration that reduces the cell viability by 50% between the top and the bottom plateaus of the obtained dose-response curve, after the normalization of the data and nonlinear regression analysis. On the other hand, the absolute value of IC50 represents the value between 0% of cell death (blank) and 100% of cell death (control value (NS) that kills 100% of the cells, usually a high concentration of the drug), which is rarely used in a pharmacological context. So, the data input in dose-response curves is the normalized values of cell viability and not the raw data obtained on MTT studies. Therefore, the value presented in Table 4 was the one adopted for all combination studies performed in this manuscript.
- In A, B, and C of figures 4 and 5, it is desirable to keep the order of the same color for the bar graphs (for the same sample condition, use the bar graph of the same color).
Our response: The figures were revised according to the reviewer suggestion.
- I suggest to reduce the concentration of 5-FU and fluphenazine to confirm the combination effect.
Our response: Thank you for your comments and suggestion. Regarding this suggestion, we would like to clarify that the current study was intended as a preliminary investigation of the potential effects of these drug combinations at the concentrations used. We chose the concentrations based on previous studies that employed similar methodologies and concentrations. However, we understand that testing lower concentrations may be beneficial in terms of reducing the occurrence of potential side effects. Therefore, in our future works, we plan to test a range of lower concentrations to investigate any potential dose-response relationships and to identify the optimal concentration that balances therapeutic effects with minimal side effects.
- In the figures 4 and 5, the doses of 5-FU and fluphenazine must be displayed, and authors also add that into figure legends.
Our response: Thank you for pointing this out. We changed Figures 4 and 5 according to the reviewer’s suggestions and added the information about the doses of 5-FU and fluphenazine into figures legends.
- Authors should add propolis in the subtitle, “2.3. Evaluation of 5-FU and fluphenazine effect on HT-29 colon cancer cell viability”.
Our response: The text was revised according to the reviewer suggestion.
- In the lines 232 and 236, uM à μM
Our response: Changes were made according to the suggestion.
Reviewer 3 Report
Comments:
1. Title should be descriptive, concise, and precise. Both 5-FU and fluphenazine were used in the study.
2. The idea of the work is published before by Sameni, H.R., Yosefi, S., Alipour, M., Pakdel, A., Torabizadeh, N., Semnani, V. and Bandegi, A.R., 2021. Co-administration of 5FU and propolis on AOM/DSS induced colorectal cancer in BALB-c mice. Life sciences, 276, p.119390, however here the authors are using different types of propolis though.
3. “3.2. Propolis samples”: please describe the region and date of collection of the red propolis.
4. Subsections 2.3, 2.4, 2.5 and 2.6: require further discussion.
5. The current study's future perspective should be included.
6. Tables 1-3 could be combined into one table to compare chemical compounds in propolis samples.
7. References should be amended to match and cover the related text (i.e. line 127: “[5,30,31], …..).
8. “In terms of total phenolic compounds, brown propolis showed the highest content” and its activity is moderate, discuss.
9. Figures 1-6: adjust the resolution and quality.
10. Some typo error should be adjusted (e,g “IC50” change to “IC50”,…..etc).
11. The authors can benefit from the following references.” Yosri, Nermeen, et al. "Anti-viral and immunomodulatory properties of propolis: Chemical diversity, pharmacological properties, preclinical and clinical applications, and in silico potential against SARS-CoV-2." Foods 10, no. 8 (2021): 1776.”
Author Response
We would like to thank the reviewers for their valuable comments, suggestions, and contributions to this study. Below you can find the specific actions regarding each comment:
Reviewer #3:
- Title should be descriptive, concise, and precise. Both 5-FU and fluphenazine were used in the study.
Our response: The title was changed considering the reviewer suggestion.
- The idea of the work is published before by Sameni, H.R., Yosefi, S., Alipour, M., Pakdel, A., Torabizadeh, N., Semnani, V. and Bandegi, A.R., 2021. Co-administration of 5FU and propolis on AOM/DSS induced colorectal cancer in BALB-c mice. Life sciences, 276, p.119390, however here the authors are using different types of propolis though.
Our response: Although some studies were previously made, our study explored different propolis samples and the most commonly found in the world market, with different botanical origin, and consequently with a distinct chemical composition. Besides we explored them using an in vitro experiment rather than using animal models.
- “3.2. Propolis samples”: please describe the region and date of collection of the red propolis.
Our response: Thanks for the reviewer comment. All the propolis samples in this study are commercial ones provided by companies. The information about the region, besides country, was not available on the product label. The collection data is mentioned on the section 3.2.
- Subsections 2.3, 2.4, 2.5 and 2.6: require further discussion.
Our response: As suggested, we improved the discussion in section 2.3 to 2.6
- The current study's future perspective should be included.
Our response: Thank you for pointing this out. We agree and have included additional information in the Conclusion section:
- Tables 1-3 could be combined into one table to compare chemical compounds in propolis samples.
Our response: It was our option to divide the tables, because the chemical profile from each type of propolis samples it presents more differences than similarities, and with such a high number of compounds in each sample, the tables design would be complex and of difficult understand for the readers.
- References should be amended to match and cover the related text (i.e. line 127: “[5,30,31], …..).
Our response: The references indicated are in line with the related sentence. In the sentence its reported that the propolis extracts analyzed had phenolic composition similar to the one described in the literature for each type of propolis, and for that we give three references were that phenolic composition is reported. The references in question are:
-Falcão, S.I.; Tomás, A.; Vale, N.; Gomes, P.; Freire, C.; Vilas-Boas, M. Phenolic quantification and botanical origin of Portuguese propolis. Ind. Crops Prod. 2013, 49, 805-812.
-Coelho, J.; Falcão, S.I.; Vale, N.; Almeida-Muradian, L.B.; Vilas-Boas, M., Phenolic composition and antioxidant activity assessment of southeastern and south Brazilian propolis, J. Apic. Res. 2017, 56 (1), 21-31.
-Vieira de Morais, D.; Rosalen, P.L.; Ikegaki, M.; de Souza Silva, A.P.; Massarioli, A.P.; de Alencar, S.M. Active Antioxidant Phenolics from Brazilian Red Propolis: An Optimization Study for Their Recovery and Identification by LC-ESI-QTOF-MS/MS. Antioxidants 2021, 10, 297.
- “In terms of total phenolic compounds, brown propolis showed the highest content” and its activity is moderate, discuss.
Our response: Although brown propolis present a higher content of phenolic compounds, the type of phenolic compounds is not the same when comparing with red and green propolis. Red propolis is mainly composed by polyprenylated benzophenones and isoflavonoids, which are described to have high biological activity. These compounds are not present in brown propolis. In that order, some amends were made in the manuscript according to the reviewer comment.
- Figures 1-6: adjust the resolution and quality.
Our response: Changes were made according to the suggestion of the reviewer.
- Some typo error should be adjusted (e,g “IC50” change to “IC50”,…..etc).
Our response: The text was revised according to the reviewer suggestion.
- The authors can benefit from the following references.” Yosri, Nermeen, et al. "Anti-viral and immunomodulatory properties of propolis: Chemical diversity, pharmacological properties, preclinical and clinical applications, and in silico potential against SARS-CoV-2." Foods 10, no. 8 (2021): 1776.”
Our response: Thanks for the suggestion.
Round 2
Reviewer 2 Report
I read carefully this revised manuscript entitled “Improvement of the in vitro cytotoxic effect on HT-29 colon cancer cells, by combining 5-fluorouacil and fluphenazine with green, red or brown propolis”. Authors made responses and corrections for the comments I had pointed out. However, this manuscript still has several problems as following:
1. About the response of comment 8:
I agree with the author’s opinion in the case of total phenolic compounds and total flavonoids, etc. However, in the case of compounds without standard substances in Tables 1 to 3, it is considered undesirable to quantify with similar compounds.
2. About the response of comment 9:
Table 1 can be included because two additional compounds have been listed, but in the case of Tables 2 and 3, if components reported from propolis of the same origin are identical with that of the currently used propolis, they should not be recorded redundantly and should be replaced with references.
3. About the response of comment 12:
In Table 4, relative IC50 was used for 5-FU and absolute IC50 was used for fluphenazine. It is undesirable to use two IC50s without distinction in one table. In addition, the IC50 value of 5-FU in the same cell line is reported in the literature as 1.43 mM (PLOS ONE | DOI:10.1371/journal.pone.0152652; April 12, 2016).
Author Response
Below you can find the specific justifications regarding each reviewer comment.
- About the response of comment 8.
I agree with the author’s opinion in the case of total phenolic compounds and total flavonoids, etc. However, in the case of compounds without standard substances in Tables 1 to 3, it is considered undesirable to quantify with similar compounds.
Our response: Thanks for the reviewer comment. Propolis, is a very complex matrix, but at the same time very rich in phenolic compounds, with more than 300 compounds already described for the phenolic extract. The discovery of new compounds within the propolis composition is frequent and every year it is possible to add several new compounds associated with propolis for the first time. These new findings are commonly made thought the study of the MS fragmentation pattern, together with the UV-Vis profile and the retention time. Other times, the confirmation of such new substances is made by NMR, due to the complexity and diversity of the chemical structures. In many cases, the comparison with commercial standards is simply not possible because they are not available in the market. Other times, although commercial available, the market value or its access is not available in all the world. For that reason, when quantifying their abundance in the propolis matrix, it is frequently to use similar compounds and express the results in equivalent terms, otherwise the quantification and the assessment of the sample composition would be reduced to few phenolic compounds, compromising an overall evaluation of the matrix. Although understanding that the final value cannot be evaluated as a absolute value, if the standards used are maintained, it would allow future comparison with other samples from different scientific studies. As mentioned in our first comment to the reviewer, this approach is being considered valid in other methodologies, including in chromatography, as ca be checked within the recent examples, between many others;
- Garcia-Oliveira, P.; Carreira-Casais, A.; Pereira, E.; Dias, M.I.; Pereira, C.; Calhelha, R.C.; Stojković, D.; Sokovic, M.; Simal-Gandara, J.; Prieto, M.A.; Caleja, C.; Barros, L. From Tradition to Health: Chemical and Bioactive Characterization of Five Traditional Plants. Molecules 2022, 27, 6495. https://doi.org/10.3390/molecules27196495
- Filaferro, M.; Codeluppi, A.; Brighenti, V.; Cimurri, F.; González-Paramás, A.M.; Santos-Buelga, C.; Bertelli, D.; Pellati, F.; Vitale, G. Disclosing the Antioxidant and Neuroprotective Activity of an Anthocyanin-Rich Extract from Sweet Cherry (Prunus avium L.) Using In Vitro and In Vivo Models. Antioxidants 2022, 11, 211. https://doi.org/10.3390/antiox11020211
- About the response of comment 9.
Table 1 can be included because two additional compounds have been listed, but in the case of Tables 2 and 3, if components reported from propolis of the same origin are identical with that of the currently used propolis, they should not be recorded redundantly and should be replaced with references.
Our response: Thank you for your comment. Within Table 1 to 3, we summarized the chromatographic, spectroscopic and mass fragmentation pattern of the compounds found in our samples, together with the quantification of each one. Although Table 2 and 3 do not introduce compounds never described for this propolis type, the information that we included in the table is not available for all the compounds, since some of them were identified in the literature with GC-MS or using other LC-MS experimental conditions that result in different fragmentation patterns. So we think that these tables are a good set of information that can useful for other future researchers, particularly those using the same chromatographic approach. Besides, we would like to call the attention to the fact that the references to previous information is given on the tables footnotes.
- About the response of comment 12.
In Table 4, relative IC50 was used for 5-FU and absolute IC50 was used for fluphenazine. It is undesirable to use two IC50s without distinction in one table. In addition, the IC50 value of 5-FU in the same cell line is reported in the literature as 1.43 mM (PLOS ONE | DOI:10.1371/journal.pone.0152652; April 12, 2016).
Our response: Thank you for your comment. We are sorry that our previous answer was not clear enough for the reviewer, but we would like to clarify that all IC50 from Table 4 were calculated in the same way and represent relative IC50s. Both IC50 values for 5-FU and fluphenazine are relative IC50 and were obtained from the dose-response curves, after the normalization of the data and nonlinear regression analysis. Particularly for fluphenazine, the relative IC50 for fluphenazine is very similar to the absolute IC50 of this drug, but we would like to reinforce that in Table 4 all IC50 were obtained following the same methodology. Regarding the IC50 value of 5-FU reported in the literature, despite it has been obtained in the same cell line, it is important to notice that this value varies with the experimental conditions such as cell density, cell media composition, treatment period, lab conditions such as temperature, etc., and for that reason we always perform the determination of the IC50 for every drug or compound in our lab. Moreover, in four previous different experiments in our group, using concentrations up to 1 mM, we found out that 5-FU reached a plateau of inhibition and that increasing its concentration would not improve its cytotoxic effect. Moreover, the important message to retain here is that we demonstrated that 5-FU, which did not have significant effects when tested alone in this cell line, when combined with propolis samples, has a potentiated cytotoxic effect, therefore demonstrating that combination treatment is better than both compounds drug alone, at concentrations that alone did not cause significant effect in the cells, but combined have been statistically significant, especially for red and brown propolis.
Reviewer 3 Report
Yes, it has been modified according to our suggestion.
Author Response
We would like to thank the reviewer for their valuable comments, suggestions, and contributions to this study. No additional suggestions were given at this stage.